# Immunogenicity Persistence of Different Immunization Regimens of Rabies Vaccine in the 10–60 Years Age Group: A Follow-Up Report Based on Phase III Clinical Trial

**DOI:** 10.3390/vaccines12111209

**Published:** 2024-10-24

**Authors:** Yunpeng Wang, Zhiang Wu, Jia Li, Shouchun Cao, Leitai Shi, Danhua Zhao, Zhaojun Mo, Haiyan Liang, Xiaohong Wu

**Affiliations:** 1Department of Biologicals Testings, National Institute for Food and Drug Control, Beijing 102629, China; wangyunpeng@nifdc.org.cn (Y.W.); lijiarv@nifdc.org.cn (J.L.); caosc@nifdc.org.cn (S.C.); taige@nifdc.org.cn (L.S.); zhaodanhua@nifdc.org.cn (D.Z.); 2Shandong Yeedo Biotechnology Co., Ltd., Dongying 257091, China; wuzhiang@yeedobio.com (Z.W.); lianghaiyan@sdydbio.com (H.L.); 3Guangxi Zhuang Autonomous Region Center for Disease Control and Prevention, Nanning 530000, China; mozhj@126.com

**Keywords:** four-dose immunization regimen, five-dose immunization regimen, immunogenicity persistence, rabies, vaccine

## Abstract

**Objectives:** This study evaluated the 12-month persistence of rabies virus-neutralizing antibody (RVNA) following different immunization regimens of freeze-dried human rabies vaccine (Vero cells) in individuals aged 10–60 years within the Chinese population. **Methods:** Number of 600 participants from phase III clinical trials who completed the fullimmunization were randomly assigned into one of three groups, four-dose experimental group, five-dose experimental group, and five-dose control group. The experimental group received the vaccine from Shandong Yeedu Biotechnology Co., while the control group used the vaccine from Liaoning Chengda Biotechnology Co., Ltd. The antibody-positive rate and geometric mean concentration (GMC) were calculated, and analysis of variance was used to compare the results among the three groups. **Results:** The study included 200 participants in the four-dose experimental group, 186 in the five-dose experimental group, and 214 in the five-dose control group. Twelve months post-immunization, the overall RVNA-positive rates were 97.00%, 93.55%, and 94.86% in the four-dose experimental, five-dose control, and five-dose experimental groups, respectively. The GMCs of RVNA were 2.50, 2.05, and 2.04 IU/mL for the respective groups. The age-stratified analysis exhibited high positivity rates across age groups (≤21, 21–50, and ≥50), with rates ranging from 88.4% to 100%, and GMCs were between 1.75 and 2.61 IU/mL. **Conclusions:** Twelve months after full immunization, the overall RVNA-positive rate remained high across all dosing regimens, demonstrating satisfactory immunogenicity persistence. This indicates that both the four-dose and five-dose regimens are effective in maintaining long-term immunity against rabies.

## 1. Introduction

Rabies is a contagious zoonotic disease caused by the rabies virus, with a mortality rate of approximately 100% [1]. There are currently no effective treatments for this condition. Unlike many other human infectious diseases, the timely administration of a rabies vaccine can prevent the development of rabies, even after exposure to the virus. Therefore, post-exposure vaccination is considered an effective method for preventing rabies disease progression [2].

Due to the higher immunogenicity of cell-culture vaccines, Europe pioneered a six-dose vaccination program with injections on days 0, 3, 7, 14, 28, and 90 [3]. In 1984, the Zagreb Institute of Public Health in former Yugoslavia developed a four-dose immunization regimen (2–1–1) [4]. The United States Advisory Committee on Immunization Practices recommended a simplified four-dose immunization program, reducing the final dose from the original five-dose regimen in 2009 [5]. The 2010 edition of “*Rabies vaccines: WHO position paper*” recommended two muscle-based post-exposure vaccination programs, the “five-dose Essen regimen” and “four-dose Zagreb regimen” [6]. Previous studies have shown that both regimens perform well in safety and immunogenicity [7,8,9].

The short-term immune effects of both vaccination procedures have been well-studied and confirmed [10], but reports on immune persistence are scarce. Studies have shown a decrease in the conversion rate of positive anti-rabies virus-neutralizing antibodies one year after full course rabies vaccination [2,7]. The immunogenicity and safety of the improved vaccine produced by Shandong Yeedo Biotechnology Co. have been evaluated in a Phase III clinical trial. The trial aimed to analyze vaccine quality under the positive trend in China’s vaccine market and to optimize vaccine production processes. At 14 days post-vaccination, the seroconversion rates were 100% in 5-dose experimental group, 5-dose control group, and 4-dose experimental group. The GMC in the 5-dose experimental, 5-dose control, and 4-dose experimental groups were 70.45 (63.02–78.77), 71.37 (63.57–80.13), and 74.30 (66.80–82.65) IU/mL. Number of 1290 AEs occurred (71.79%), with incidence rates of 72.79%, 70.40%, and 72.17% in the 5-dose experimental, 5-dose control, and 4-dose experimental groups, respectively. The data from our Phase III clinical trial indicated that our vaccine has demonstrated good performance in immunogenicity and safety.

Given that studies and evidence on vaccine immune persistence are still limited [2,7,11], we conducted a 12-month immune persistence study in connection with our Phase III clinical trial. Shi et al. [2] found that 365 days after vaccination with the Liaoning Chengda vaccine, antibody levels remained high, with rates over 75% for the Zagreb regimen and over 50% for the Essen regimen. Ma et al. [7] reported 13 months after Novartis Vaccines, GMCs were 2.8 IU/mL (Zagreb) and 2.7 IU/mL (Essen), with seroconversion rates of 90.4% and 94.8%. Zhang et al. [11] noted that one year after the Liaoning Chengda vaccine, GMTs stayed protective (>0.5 IU/mL), though seroconversion rates fell below 80%. The aim of this study was to evaluate the immune persistence of a purified Vero cell-based rabies vaccine produced by Shandong Yeedo Biotechnology Co. 12 months after full vaccination in a five-dose Essen regimen and a four-dose Zagreb regimen. Our vaccine demonstrates improved immune durability through process optimization, such as the use of bioreactors combined with microcarriers perfusion culture technology. We hypothesized that the vaccine would maintain a high seroconversion rate after a 12-month follow-up.

## 2. Materials and Methods

### 2.1. Patients

This study is an analysis of immune persistence based on a Phase III clinical trial of a freeze-dried human rabies vaccine (Vero cells) (Registration No.: CTR20182016, http://www.chinadrugtrials.org.cn/index.html, accessed on 10 August 2024).

The trial included Guangxi Province residents aged 10–60 years who had not been vaccinated against rabies, had not received any other vaccines, antisera, human immunoglobulins, or similar products within the month prior to enrollment. We excluded those with a history of severe allergy to vaccines, as well as patients with severe cardiovascular diseases, liver or kidney diseases, and mental illnesses. A total of 1800 participants were enrolled in the Phase III clinical trial and randomized into three groups in a 1:1:1 ratio, with 600 participants in each group. Recruitment of participants began in March 2018 based on informed consent and voluntary participation principles. Blood sample collection lasted until May 2019, covering the entire immunization process and 12 months post-immunization. This study subsequently enrolled patients sequentially into the four-dose experimental, five-dose experimental, and five-dose control groups.

### 2.2. Intervention

The experimental vaccine was Vero cells produced by Shandong Yeedo Biotechnology Co., Ltd. (Dongying, China) (0.5 mL per vial after reconstitution, with a potency of rabies vaccine not less than 2.5 IU), batch number T20170901. The control vaccine was Vero cells produced by Liaoning Chengda Biotechnology Co., Ltd. (Shenyang, China) (0.5 mL per vial), batch number 201704110. Liaoning Chengda was selected as the control vaccine, considering the same cell matrix and virulence for production and close production process [12,13]. All vaccines were transported and stored at 2–8 °C. Randomization codes were generated using the SAS 9.4 software for the allocation and numbering of experimental and control vaccines. The five-dose experimental group and five-dose control group received five doses of the experimental or control vaccine at 0, 3, 7, 14, and 28 days, while the four-dose experimental group received four doses of the experimental vaccine at 0, 7, and 21 days (two doses injected on day 0 in both arms).

### 2.3. Sample Testing

Blood samples were collected for rabies specific antibody detection from all patients before the first immunization, 7 days after the first immunization, 14 days after the first immunization, and on the 14th day after completion of the full immunization. Blood was collected 12 months after the completion of full immunization (window period: +2 months) to assess immune persistence.

The immunogenicity persistence study used the Phase III clinical trial research numbers as unique identifiers, with only the blood sample numbers reassigned. In this study, a new blood collection number was assigned for identification, and the blood collection number was C001–C600. For the four-dose experimental group, the first 200 subjects meeting the inclusion and exclusion criteria for observation of immune persistence were enrolled according to the order of visit, and the blood collection number will be C001–C200. Since our Phase III clinical trial was still blinded before the initiation of this trial, we had no information if each participant was enrolled to the five-dose control group or five-dose experimental group; therefore, in the five-dose program group, continuous numbers were selected as far as possible to ensure that the numbers of the experimental group and the control group were similar. The first 400 subjects meeting the inclusion and exclusion criteria for observation of immune persistence were enrolled according to the order of this visit, and blood collection numbers were C201–C600. Due to differences in immunization and blood sampling time, blinding was not applied to the Essen regimen (five-dose groups) and Zagreb regimen (four-dose group). In the five-dose experimental and five-dose control groups, the subjects, researchers, and statisticians were blinded.

Approximately 3.0 mL of venous blood was collected and left at room temperature for 1–2 h To allow serum stratification, after which they were centrifuged. If immediate centrifugation was not possible, the serum was stored in a refrigerator at 4–8 °C and separated within 24 h. The separated serum was divided into two parts: one part was used for serum antibody detection (not less than 0.5 mL), and the other was kept as a spare, with priority given to meeting the testing requirements. The serum was stored at −20 °C or below. The rapid fluorescent focus inhibition test was used to detect rabies vaccine virus-neutralizing antibodies in the serum, with an antibody concentration of ≥0.5 IU/mL considered positive. The serum antibody positivity rate and geometric mean concentration (GMC) were calculated for the three groups 12 months after the completion of full immunization.

### 2.4. Evaluation of Immune Persistence

The antibody positivity rate and antibody GMC at 12 months after the full immunization series were used for evaluation. Antibody positivity was defined as participant pre-immunization antibody concentrations of <0.5 IU/mL and post-immunization antibody concentrations ≥0.5 IU/mL. The positivity rate is defined as the proportion of seropositive cases.

### 2.5. Statistical Analysis

According to the Chinese Guidelines for Clinical Research on Human Rabies Vaccines [14], 600 participants from the Phase III clinical trial were selected, with 200 in the four-dose experimental group and 400 in the combined five-dose experimental and five-dose control groups. The Phase III clinical trial was still blinded, consecutive number ranges were used to ensure similar numbers of participants in the five-dose experimental and five-dose control groups. All statistical analyses were performed using the SAS 9.4 software. The analysis set at 12 months after the full immunization series (IPS-12) included all participants who entered the immune persistence evaluation, completed blood collection 12 months after full immunization series, and had valid antibody values. Descriptive statistics were used for continuous data, including means and standard deviations, and categorical data, including frequencies and percentages. The Clopper–Pearson method was used to calculate the 95% confidence interval for the serum antibody positivity rate 12 months after full immunization series for each group, and the chi-square test or Fisher’s exact probability test was used to statistically analyze the differences between groups. Geometric means and 95% confidence intervals were used to describe the antibody GMC at 12 months after full immunization series for each group and the fold increase compared to that at 14 days after full immunization series. The fold increase is defined as the ratio of the 12-month post-immunization GMC to the pre-vaccination GMC. Statistical analysis of differences between groups was performed using analysis of variance with log-transformed data. Reverse distribution plots of antibody concentration at pre-immunization, 7 days after the first immunization, 14 days after the first immunization, 14 days after full immunization series, and 12 months after full immunization series as well as antibody concentration-time semi-log plots before and after immunization were generated for each group. Subgroup analysis was conducted based on pre-immunization-positive and pre-immunization-negative participants. Missing data from immune persistence endpoints were not processed in this trial to maintain data originality and authenticity.

### 2.6. Ethical Review

Prior to the start of this study, it was approved by the Ethics Review Committee of the Guangxi Zhuang Autonomous Region Center for Disease Control and Prevention. All participants or legal guardians were fully informed about the relevant matters of the clinical trial and provided voluntary informed consent.

## 3. Results

### 3.1. Baseline Characteristics

A total of 600 patients were enrolled: 200 in the four-dose experimental group, 214 in the five-dose experimental group, and 186 in the five-dose control group. All the patients were included in the IPS-12 analysis set (Table 1 and Figure 1). There were no statistically significant differences in age or sex among the patients (*p* > 0.05).

### 3.2. Immunological Characteristics Analysis

A comparison of pre-immune antibody positivity rates and GMC among the groups is shown in Table 2 and Appendix A. The antibody positivity rates 7 days after the first dose were 79.00%, 46.07%, and 59.46% in the four-dose experimental, five-dose experimental, and five-dose control groups, respectively, which were statistically different among the three groups (*p* < 0.001). The antibody positivity rate 14 days after the first dose immunization and 14 days post-vaccination was 100% in all three groups, but the difference was not statistically significant (*p* > 0.05).

### 3.3. Analysis of Immunogenic Persistence

Twelve months post-immunization, the overall antibody positivity rates were 97.00%, 93.55%, and 94.86% in the four-dose experimental, five-dose control, and five-dose experimental groups, respectively, with no statistically significant differences among the three groups (*p* = 0.277). The overall antibody GMCs at 12 months post-immunization were 2.50 IU/mL, 2.05 IU/mL, and 2.04 IU/mL in the four-dose experimental, five-dose control, and five-dose experimental groups, respectively, with no statistically significant differences among the groups (*p* = 0.184). The fold increases in overall antibody GMCs at 12 months post-immunization compared with 14 days post-immunization were 0.05, 0.05, and 0.04 in the four-dose experimental, five-dose control, and five-dose experimental groups, respectively, with no statistically significant differences among the three groups (*p* = 0.421) (Table 3).

Table 4 presents an age-stratified analysis of antibody positivity rates and GMC 12 months post-immunization. In the four-dose experimental group, positivity rates were 98.0% (≤21), 97.1% (21–50), and 95.6% (≥50), with no significant difference across ages. The five-dose control group exhibited slightly lower rates: 93.8% (≤21), 94.7% (21–50), and 90.0% (≥50), also with no significant intra-group difference. The five-dose experimental group had the highest rates in the ≤21 group (100.0%), but the positivity decreased in older participants, especially for those ≥50 years (88.4%) (Pintra = 0.055). The GMCs were similar across all age groups and groups, ranging from 1.75 to 2.61 IU/mL. No significant differences in GMCs were observed within any age group or between groups (all P_intra_ > 0.05). There was no statistical difference between the same age groups.

### 3.4. Distribution of Antibody Levels at 12 Months Post-Immunization

The reverse distribution table of overall antibody concentrations at 12 months post-immunization indicates that all groups were able to effectively produce neutralizing antibodies in participants positive for antibodies (neutralizing antibody concentration ≥0.5 IU/mL), with no significant differences among groups, except neutralizing antibody concentration ≥ 4 (Table 5).

### 3.5. Subgroup Analysis

Among pre-immune positive participants (antibody concentration ≥0.5 IU/mL), the GMCs remained at relatively high levels at 12 months post-immunization, with GMCs of 5.71 IU/mL, 8.24 IU/mL, and 6.15 IU/mL in the four-dose experimental, five-dose control, and five-dose experimental groups, respectively, with no statistically significant differences among the three groups (*p* = 0.771). In pre-immunity-negative participants, the positivity rates at 12 months post-immunization were above 85% (88.0%, 87.6%, and 86.9%, respectively), with no statistically significant differences among the groups (*p* = 0.308) (Table 6).

## 4. Discussion

This study compared and analyzed the immunogenicity of four-dose experimental, five-dose experimental, and five-dose control groups as an extension of a phase III clinical trial. The results showed that, after full immunization for 12 months, the overall antibody positivity rates of the four-dose experimental, five-dose control, and five-dose experimental groups, were slightly lower than those 14 days after full immunization, but remained at a relatively high level (97.00% for the four-dose experimental, 93.55% for the five-dose control, and 94.86% for the five-dose experimental groups).

Currently, the main immunization programs in China are the five-dose Essen scheme and the four-dose Zagreb scheme, both of which have good immunogenicity and safety [2,4]. The study results indicate that over time, the protective level of neutralizing antibodies gradually decreases, but remains at a relatively high level after reaching its peak 14 days after the first immunization. Notably, all three groups were able to maintain a positivity rate of over 90% after 12 months of vaccination, which is much higher than the antibody positivity rate of participants in previous studies after one year [2,15] and similar to the antibody positivity rate after one year reported by Cramer et al. [16]. In age-based stratification analysis, ≤21 age group had higher positivity rate (98.0% and 100%) in four-dose experimental group and five-dose experimental group and higher GMC (2.18 IU/mL) in five-dose experimental group among three age groups. ≥50 age group had higer GMC (2.61 IU/mL) in four-dose experimental group. 21–50 age group had higher positivity rate (94.7%) and ≥50 age group had higher GMC (2.08 IU/mL) in five-dose control group. ≤21 and ≥50 groups might have better response to 4-dose or 5-dose vaccine program. The present study also showed that in the subgroup analysis, antibody levels in seronegative individuals before vaccination remained high, with a positivity rate of over 85%. This trend suggests that the vaccine generated an effective immune response after initial immunization, providing recipients with good protection and long-lasting immunity. This sustained immune effect may be related to the specificity of vaccine design and immunization programs.

Although no statistically significant differences were observed in GMC levels among the three groups during the 12-month follow-up period after immunization the GMC levels in all groups remained above 0.5 IU/mL, indicating antibody positivity. This suggests that all vaccination regimens can maintain sufficiently high levels of antibodies, ensuring effective protection against the rabies virus, even in the absence of statistical differences. Notably, the GMC antibody in the four-dose experimental group was considerably higher than those in the five-dose experimental group. This may indicate that with this specific vaccine regimen, lower doses can generate a stronger and more durable immune response, possibly related to factors, such as immunological memory and response intensity [17]. However, in Zhang et al.‘s research [18], the seroconversion rate of Liaoning Chengda vaccine gradually declined from 90.5% at one year to 34.0% at five years, suggesting that the durability of antibodies in this study’s experimental vaccine warrants investigation through a longer follow-up.

In addition, the antibody positivity rate of all three groups did not reach 100% after one year, which is consistent with the results of previous studies [1], suggesting that WHO-recommended immunization procedures should be followed [6] to achieve protection. Booster vaccinations were administered six months after full immunization, if necessary.

The strengths of this study include improvements to the vaccine production process, such as using advanced microcarrier bioreactors combined with microcarrier perfusion culture technology to ensure vaccine quality and safety and using single-use sterile bags for aseptic transfer from the bioreactor to the lyophilized formulation. After process optimization, the seroconversion rate after 12 months remains higher than in previous studies [2,7]. Following the enactment of the National Law on Vaccine Management [19], which provides a legal framework for vaccine clinical trials. Our vaccine demonstrates improved immune durability through process optimization, such as the use of bioreactors combined with microcarriers perfusion culture technology. The results are superior compared to previous studies using control vaccines, providing new evidence for the long-term immunogenicity of vaccines in China.

This study has some limitations. One limitation of this study is the relatively short follow-up period of 12 months, which may not fully capture the long-term durability of immunity provided by the rabies vaccines. This timeframe limits our ability to assess how immune responses evolve over time and may overlook potential waning immunity beyond the observed period. Compliance was not analyzed in this study. According to a previous report [20], compliance with the four-dose Zagreb regimen was markedly higher than that with the five-dose Essen regimen, possibly due to patients forgetting vaccination times or inability to adhere to the specified vaccination schedule due to work or study commitments. Future research should include long-term compliance analyses for both vaccination regimens. This study lacks the collection of safety data. It is important to clarify that determining the correlation (causality) between adverse effects within 12 months post-immunization and the study vaccine is challenging. We will certainly pay more attention to the collection of adverse events in future studies and will be more cautious and thorough in assessing the correlation between adverse events and the study vaccine.

## 5. Conclusions

Both four-dose and five-dose vaccination regimens can induce sustained immune memory and have good immunological durability within 12 months, effectively immunizing the body against the rabies virus. The results of this study provide a basis for the selection and refinement of various immunization programs. In future, we will focus on long-term follow-up studies and post-exposure booster immunization research. We will conduct stratified analyses for different age groups to optimize vaccination strategies. In addition, we will evaluate the cost-effectiveness of different vaccination regimens to guide public health policy formulation.

## Figures and Tables

**Figure 1 vaccines-12-01209-f001:**
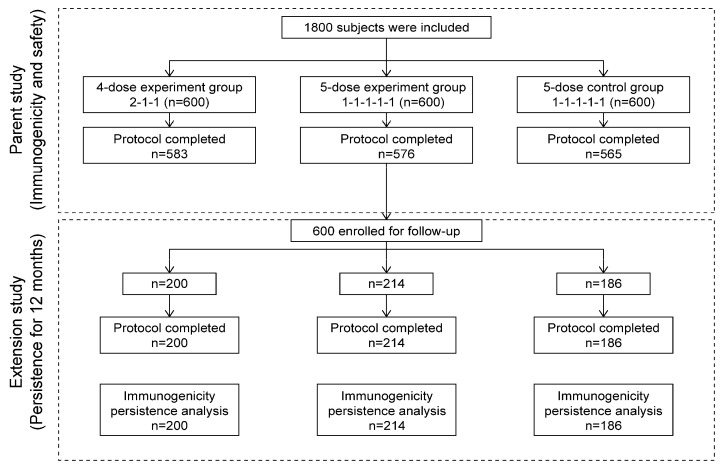
Study flowchart.

**Table 1 vaccines-12-01209-t001:** Participants’ characteristics.

	Four-Dose Experimental Group (n = 200)	Five-Dose Control Group (n = 186)	Five-Dose Experimental Group (n = 214)	*p* Value
Age, mean ± SD (median)	36.55 ± 15.44 (40.67)	39.67 ± 13.83 (43.93)	38.70 ± 13.60 (41.89)	0.089
Gender (male:female)	83:117	67:119	73:141	0.277

**Table 2 vaccines-12-01209-t002:** Immunological Characteristics Analysis.

Items	Four-Dose Experimental Group (n = 200)	Five-Dose Control Group (n = 186)	Five-Dose Experimental Group (n = 214)	*p* Value
**Pre-immune**				
Antibody positivity rates, %	9.00 (5.42–13.85)	5.91 (2.99–10.34)	8.41(5.06–12.97)	0.490
GMC (IU/mL)	0.10 (0.09–0.12)	0.09 (0.08–0.10)	0.10 (0.08–0.11)	0.708
**Day 7 post-first dose**				
Antibody positivity rate, %	79.00 (69.71–86.51)	46.07 (35.44–56.96)	59.46 (49.73–68.68)	<0.001
GMC (IU/mL)	1.34 (0.94–1.90)	0.40 (0.30–0.54)	0.55 (0.37–0.80)	<0.001
**Day 14 post-first dose**				
Antibody positivity rate, %	100.00 (96.38–100.0)	100.00 (96.27–100.0)	100.00 (96.48–100.0)	>0.999
GMC (IU/mL)	80.02 (66.02–96.99)	85.49 (69.58–105.05)	78.19 (64.42–94.90)	0.807
**Overall 14 days post-vaccination**				
Antibody positivity rate, %	100.00 (98.17–100.00)	100.00 (98.04–100.00)	100.00 (98.29–100.00)	>0.999
GMC (IU/mL)	49.28 (42.73–56.83)	44.56 (38.35–51.78)	46.37 (40.47–53.12)	0.621

GMC means Geometric Mean Concentration.

**Table 3 vaccines-12-01209-t003:** Antibody positivity rates and GMC of 12 months post-immunization.

	Four-Dose Experimental Group (n = 200)	Five-Dose Control Group (n = 186)	Five-Dose Experimental Group (n = 214)	*p* Value
Number of positivity	194	174	203	
Positivity rate (95% CI)	97.00 (93.58–98.89)	93.55 (89.00–96.62)	94.86 (90.99–97.41)	0.277
GMC (IU/mL)	2.50 (2.12–2.94)	2.05 (1.69–2.49)	2.04 (1.73–2.42)	0.184
GMC increase time	0.05 (0.04–0.06)	0.05 (0.04–0.05)	0.04 (0.04–0.05)	0.421

GMC means Geometric Mean Concentration.

**Table 4 vaccines-12-01209-t004:** Age-stratified analysis antibody positivity rates and GMC of 12 months post-immunization.

	Four-Dose Experimental Group (n = 200)				Five-Dose Control Group (n = 186)				Five-Dose Experimental Group (n = 214)			
Age (Year)	≤21	21–50	≥50	P_intra_	≤21	21–50	≥50	P_intra_	≤21	21–50	≥50	P_intra_
Number of positivity	49	102	43		30	108	36		36	129	38	
Positivity rate (95% CI)	98.0 (94.12, 100.00)	97.1 (93.96, 100.00)	95.6 (89.53, 100.00)	0.778	93.8 (85.36, 100.00)	94.7 (90.64, 98.84)	90.0 (80.70, 99.30)	0.576	100.0 (100.00, 100.00)	95.6 (92.08, 99.03)	88.4 (78.79, 97.95)	0.055
GMC (IU/mL)	2.34	2.53	2.61	0.894	1.95	2.07	2.08	0.971	2.18	2.11	1.75	0.660

**Table 5 vaccines-12-01209-t005:** Reverse distribution of overall antibody concentrations at 12 months post-immunization.

Neutralizing Antibody Concentration (IU/mL)	Four-Dose Experimental Group (n = 200)		Five-Dose Control Group (n = 186)		Five-Dose Experimental Group (n = 214)		*p*
	Number	%	Number	%	Number	%	
≥0.5	194	97.00	174	93.55	203	94.86	0.2774
≥1	165	82.50	139	74.73	160	74.77	0.1018
≥2	92	46.00	64	34.41	84	39.25	0.0648
≥4	74	37.00	49	26.34	58	27.10	0.0355
≥8	35	17.50	29	15.59	28	13.08	0.4569
≥16	10	5.00	16	8.60	14	6.54	0.3646
≥32	6	3.00	9	4.84	9	4.21	0.6424
≥64	0	0.00	4	2.15	2	0.93	0.0861
≥128	0	0.00	2	1.08	0	0.93	0.4707
≥256	0	0.00	0	0.00	0	0.00	1.0000

**Table 6 vaccines-12-01209-t006:** Immunological characteristics of pre-immune positive and pre-immune negative participants.

	Four-Dose Experimental Group (n = 200)	Five-Dose Control Group (n = 186)	Five-Dose Experimental Group (n = 214)	*p*
Number of pre-immune positive participants, n	18	11	18	
Pre-immune GMC (IU/mL)	1.32	1.27	1.45	0.933
12 months post-immunization GMC (IU/mL)	5.71	8.24	6.15	0.771
12 months post-immunization positivity rate, %	18 (9.00%)	11 (5.91%)	17 (7.94%)	0.439
Number of pre-immune negative participants, n	182	175	196	
Pre-immune GMC (IU/mL)	0.08	0.08	0.07	0.861
12 months post-immunization GMC (IU/mL)	2.30	1.88	1.85	0.148
12 months post-immunization positivity rate, %	176 (88.00%)	163 (87.63%)	186 (86.92%)	0.308

## Data Availability

The data that support the findings of this study are available from Shandong Yeedo Biotechnology Co., Ltd. but restrictions apply to the availability of these data, which were used under license for the current study, and so are not publicly available. Data are, however, available from the authors upon reasonable request and with permission of Shandong Yeedo Biotechnology Co., Ltd.

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
