# Peer review of "Immunogenicity Persistence of Different Immunization Regimens of Rabies Vaccine in the 10–60 Years Age Group: A Follow-Up Report Based on Phase III Clinical Trial"

_vaccines, 2024, doi:10.3390/vaccines12111209_

Round 1

Reviewer 1 Report

Comments and Suggestions for Authors

Abstract

The abstract should be prepared as a single structured paragraph. 

It is not clear which one is the objective.

It is not clear how the groups were defined. For instance, why the control group was a five-dose. What was the difference between the five-dose trial group and the five-dose control group, this should be straightforward to understand. 

The results are not easy to understand.

The sample size should be included in the methods. 

Given that the methods and results are unclear, the conclusions are not supported by the results.

There is a potential conflict of interest because one of the authors works in the company where the vaccine was developed. The manuscript does not declare this COI. Besides, data to reproduce the study is unavailable because the company restricts it. Accordingly, how can a reader be sure that the data from which the results were obtained are accurate and real? 

Introduction

This section was poorly prepared and developed, the rationale is weak and lacks convincing arguments. 

"The immunogenicity and safety of the improved vaccine produced by Shandong Yeedu Biotechnology Co. have been evaluated in a Phase III clinical trial. The trial aimed to analyze vaccine quality under the positive trend in China's vaccine market and to optimize vaccine production processes. The data from our Phase III clinical trial indicated that our vaccine has demonstrated good performance in immunogenicity and safety." these statements are vague and are not supported by data, results or previous publications. Therefore, they lack solid scientific arguments. 

"Given that studies and evidence on vaccine immune persistence are still limited" There is no other argument to support this vague statement. Again, the rationale for performing the study is weak. 

"We hypothesized that the vaccine would maintain a high seroconversion rate after a 12-month follow-uo" What was the empirical data or evidence to support such a statement? The authors rely solely on their knowledge and experience, but without presenting convincing arguments and data. 

Due to the listed major issues, I recommend that this manuscript should be rejected for possible publication in the journal. 

Reviewer 2 Report

Comments and Suggestions for Authors

Wang and colleagues report here the 12 month follow up report based on Phase III Clinical Trial. As highlighted by the authors, very few long-term follow up studies have been carried out for vaccinees in clinical trials. As part of the primary finding, the authors report no significant differences between recipients of 4-dose or 5-dose (control and test) regiments. All three vaccine schedules demonstrated good prolonged immunogenicity. This information is useful within the field and would help with further optimization of vaccine plans especially targeting zoonotic diseases such as Rabies.

Comments to the authors:

1. Section 2.3: It would be good to include the kit (Kit Manufacturer) that was used and any further conditions used to calculate the GMC.

2. The data in Table 2 can be consolidated as a line graph for better visualization for readers in addition to the information provided in the table.

3. Were there any age-based differences observed with regards to the 12-month post immunization GMC values? - Which age groups showed better response to 4-dose or 5-dose vaccine program?

Reviewer 3 Report

Comments and Suggestions for Authors

Overall, this is a well written manuscript that provides important findings on rabies vaccination protocols and persistence of immunity.  It is encouraging to know that the vaccines were effective in seroconverting individuals to a protected status.  The points raised below should help to clarify issues or assist the reader in clarifying concepts of the study.

Lines 17-19 There should be a quick description that the “control” group is in fact a different vaccine control, and not what people expect to see based on a “no treatment” control.  Also, add a phrase in this section that states that the participants were randomly assigned into one of three groups.

 Line 40:  suggested wording “…post-exposure vaccination is considered an effecteve method for preventing rabies disease progression”

Line 54 suggested wording “…one year after full course rabies vaccination”

Line 73-75 suggested wording “…had not received any other vaccine, antisera, human immunoglobulins, or similar products within the month prior to enrollment.  We excluded those with…”

Line 97 suggested wording “Blood samples were collected for rabies specific antibody detection from all patients…”. Eliminate the phrase antibody detection at end of that sentence. 

Line 105 – there is an indication that some of the study was blinded and some of it was not, specifically with the five-dose regimens.  Can this be better explained why that was the case in the methods section?

Line 107 – I understand that there was use of original designation for individual subjects in the four-dose study, that carried over to the antibody longevity analysis at one year, but I do not understand still the reason why the second two arms of the study could not carry over their originally assigned study ID code, if I am interpreting this correctly.

Lines 125-127 – There appear to be a possible study inclusion/exclusion criterion of a pre-vaccination titer that was <0.5 IU/ml, but in Table 5, there is an analysis of individuals from each of the study arms who already had titers above this cutoff level.  Is that correct?  If so, I assume there was no testing to exclude individuals, but perhaps these were identified after vaccinations had begun.  Is that the correct interpretation?  What would happen if those individuals were removed from the analysis?

Line 138 – there is a statement for the inclusion parameter IPS-12 that suggests that you may have removed individuals who did not “have valid antibody values”.  How many, if any, individuals might that have been and what impact might that have on the analysis between regimens?

Line 149 – suggested wording “…after the full immunization series..”

Line 162-163 – I am unsure why there was a breakdown of 200 subjects in one group, and then 214 and 186 in the next two groups.  Why was the design not for 200 in each group?  Was there a reason for that unusual study number per group?

Figure 1 – study flow chart – what is the explanation for starting with 1800 subjects and then the follow up with 600.  Was this the reason above for their being numerical differences in the 600 person long-term study, i.e. refusal to participate, lost to follow-up, etc.?  I think that was somewhat explained but it is still unclear.

Line 186-187 – I am having trouble understanding the notion of the fold increase.  Does 0.05 for instance mean that the fold increase was 5% (as in 1.05x) or does it mean that the second screening was lower based on concentration (GMT) at the endpoint measure?  I could understand the sentence if you were to say that there was a 25% increase (or decrease) in the two time points.

Line 257 – the phrase “…this study offers new evidence for vaccine research in China” is vague.  Do you mean that it connects to the impact of the changed vaccine production method mentioned between the two vaccines, or is there some other meaning about conduct of studies with vaccine in China?

Reviewer 4 Report

Comments and Suggestions for Authors

The present manuscript entitled "Immunogenicity Persistence of Different Immunization Regimens of Rabies Vaccine in the 10–60 Years Age Group: a Follow-up Report Based on Phase III Clinical Trial" addresses an important public health issue by evaluating the long-term persistence of rabies vaccine immunity. Rabies is fatal, and the research provides important data on the durability of immunogenicity for rabies vaccines.

However, the authors need to address some important points:

1. While this study provides valuable insights into the 12-month immune persistence, the follow-up period is relatively short. Rabies vaccines are critical for long-term immunity, and a more extended follow-up (from 3 to 5 years) would provide a better understanding of the durability of immunity.

2. The study includes participants aged 10–60 years but does not perform an age-stratified analysis. Immune responses can vary significantly across different age groups, particularly between children and older adults.

3. The authors need to address safety outcomes in more detail, focusing on  long-term adverse effects that might arise from different immunization schedules.

4. The study is funded by Shandong Yeedo Biotechnology Co., Ltd., which also produces the experimental vaccine tested in the trial. The funding source could introduce bias, especially in the interpretation of the results. Independent verification of the findings is necessary to confirm their validity.

Round 2

Reviewer 1 Report

Comments and Suggestions for Authors

Confidential comments to the Editor. 

Reviewer 4 Report

Comments and Suggestions for Authors

The authors have responded to my comments.